# Bioactive Phenolic Metabolites from Adriatic Brown Algae *Dictyota dichotoma* and *Padina pavonica* (Dictyotaceae)

**DOI:** 10.3390/foods10061187

**Published:** 2021-05-25

**Authors:** Ivana Generalić Mekinić, Vida Šimat, Viktorija Botić, Anita Crnjac, Marina Smoljo, Barbara Soldo, Ivica Ljubenkov, Martina Čagalj, Danijela Skroza

**Affiliations:** 1Department of Food Technology and Biotechnology, Faculty of Chemistry and Technology, University of Split, R. Boškovića 35, HR-21000 Split, Croatia; viktorijabotic9@gmail.com (V.B.); anitaaklanac@gmail.com (A.C.); marinasmoljo.1@gmail.com (M.S.); danci@ktf-split.hr (D.S.); 2Department of Marine Studies, University of Split, R. Boškovića 37, HR-21000 Split, Croatia; martina.cagalj@unist.hr; 3Department of Chemistry, Faculty of Science, University of Split, R. Boškovića 33, HR-21000 Split, Croatia; barbara@pmfst.hr (B.S.); ivica.ljubenkov4@gmail.com (I.L.)

**Keywords:** antioxidants, biological activity, brown seaweeds, extraction mode, phenolic compounds

## Abstract

In this study, the influences of temperature (20, 40 and 60 °C) and extraction solvents (water, ethanol) on the ultrasound-assisted extraction of phenolics from the Adriatic macroalgae *Dictyota dichotoma* and *Padina pavonica* were studied. The extracts were analysed for major phenolic sub-groups (total phenolics, flavonoids and tannins) using spectrometric methods, while the individual phenolics were detected by HPLC. The antioxidant activities were evaluated using three methods: Ferric Reducing/Antioxidant Power (FRAP), scavenging of the stabile 2,2-diphenyl-1-picrylhydrazyl (DPPH) radical and Oxygen Radical Antioxidant Capacity (ORAC). The aim of the study was also to find the connection between the chemical composition of the extracts and their biological activity. Therefore, principal component analysis (PCA), which permits simple representation of different sample data and better visualisation of their correlations, was used. Higher extraction yields of the total phenolics, flavonoids and tannins were obtained using an alcoholic solvent, while a general conclusion about the applied temperature was not established. These extracts also showed good antioxidant activity, especially *D. dichotoma* extracts, with high reducing capacity (690–792 mM TE) and ORAC values (38.7–40.8 mM TE in 400-fold diluted extracts). The PCA pointed out the significant influence of flavonoids and tannins on the investigated properties. The results of this investigation could be interesting for future studies dealing with the application of these two algae in foods, cosmetics and pharmaceuticals.

## 1. Introduction

Macroalgae (seaweeds), which are organisms that inhabit marine or freshwater habitats, have attracted the interest of researchers as they are a proven source of various biologically active metabolites. These components are generated as a protection against free radicals or other oxidizing substances that affect macroalgae since they are exposed to adverse environmental conditions [1,2,3,4,5].

Based on their photosynthetic pigment type, type of storage material and composition of cell wall polysaccharides, macroalgae are usually divided into three groups: green, red and brown algae [6,7]. Brown algae constitute the largest class (with up to 2000 species) [8] and are probably the most investigated, as these species contain a special group of biologically active phenolics called phlorotannins [7,9]. The phenolic compounds present in brown algae species are derived from polymerised phloroglucinol units via the acetate malonate pathway [10]. Phlorotannins play a role in protecting algae against herbivores, bacteria and fouling organisms. They are also involved in protection against oxidative damage, as their antioxidant activity is 10 to 100 times more powerful than that of other polyphenols [11,12]. They have also been broadly investigated since they have a large spectrum of positive biological properties such as UV protective actions and anti-inflammatory, anti-angiogenic, anti-allergic and antidiabetic effects [13,14]. Phlorotannins exist in soluble or in cell-wall-bound forms. Based on the number and distribution of hydroxyl groups and the nature of the structural linkages between phloroglucinol units, they can be divided into phlorethols, fuhalols, fucols, fucophlorethols and eckols [5,12,15,16]. Seaweeds are also an excellent source of other biologically active compounds such as polysaccharides (fucoidan, laminarin and alginates), peptides, polyunsaturated fatty acids, pigments (carotenoids), minerals, sterols, etc. [17,18].

In recent decades, great attention has been devoted to the search for new, bioactive-compound-rich and inexpensive natural sources of valuable phytochemicals with beneficial biological activities. Seaweeds from diverse habitats have great potential due to their valuable nutritional profile, low caloric value and medicinal benefits. They are often used as vegetables (fresh or dried) and/or ingredients in numerous dishes, and they are also often used as valuable ingredients in the formulation of functional foods [17,18,19]. The brown macroalgae *Dictyota dichotoma* and *Padina pavonica* (Dictyotaceae) are species widespread in the Adriatic Sea, but there are limited or scarce data on their chemical composition [20,21], especially their phenolic content [22]. Due to the long duration and application of high temperatures in conventional extraction protocols, negative effects on their health-promoting components are often observed. Therefore, there is an increasing trend toward the investigation of novel, environmentally friendly extraction technologies for the preparation of highly valuable extracts. Among the different new technologies, the use of ultrasound-assisted extraction is probably one of the easiest, cheapest and, consequently, most widely used [5,15,18,23]. 

The aim of this study was to investigate the biological potential of *D. dichotoma* and *P. pavonica* and the impact of the extraction mode (ultrasound, solvent and temperature regime) on their phenolic profile. Furthermore, the connection between the phenolic profile of the algae and related activity was investigated in order to find the extraction parameters best suited for producing bioactive extracts. 

## 2. Materials and Methods

### 2.1. General

All chemicals, reagents and solvents used were of adequate analytical grade and were obtained from Kemika (Zagreb, Croatia) and Sigma-Aldrich (St. Louis, MO, USA). Spectrophotometric measurements were performed on a SPECORD 200 Plus, Edition 2010 (Analytik Jena AG, Jena, Germany) and a Synergy HTX Multi-Mode Reader (BioTek Instruments, Inc., Winooski, VT, USA). The high-performance liquid chromatography (HPLC) system used was the Perkin Elmer Series 200 with a UV/VIS detector (Perkin-Elmer Inc., Shelton, CT, USA), and the phenolic compounds were separated on an UltraAqueous column (C18, 250 × 4.6 mm, 5 mm, Restek, Bellefonte, PA, USA).

### 2.2. Seaweed Material and Preparation of Extracts

Seaweed materials, *Dictyota dichotoma* (Hudson) J. V. Lamouroux and *Padina pavonica* (L.) Gaill. were collected in August 2020 on the coast of Čiovo Island (Central Dalmatia, Croatia, 43.523492° N, 16.285571° E). Zvjezdana Popović Perković, a marine botanist from the University of Split’s Department of Marine Studies, confirmed the botanical identity of the algal materials. The algal biomass was harvested by hand and then washed thoroughly with fresh water to remove epiphytes. The materials (algae thalli) were air-dried for 15 days in a shaded and aerated place at room temperature, and then, they were pulverised (1 min in high-speed grinder, Model 980, Moulinex, France) and used for the preparation of extracts.

The algal extracts were prepared by ultrasound-assisted extraction (Sonorex RK 103H, Bandelin, Berlin, Germany) using different solvents: ethanol and water at three different temperatures of 20 (room temperature, RT), 40 and 60 °C. For algal extracts, 1 g of dried and ground samples was weighed and 10 mL of solvent was added. The duration of the extraction was 1 h, following which the suspensions were filtered and centrifuged (10 min, 4000 rpm, Centric 322 A, Tehtnica, Slovenia). Extractions were carried out in three repetitions for each plant material, and all sample extracts were combined in a total extract that was used for further analysis.

### 2.3. Phenolic Composition

#### 2.3.1. Spectrophotometric Analysis of Phenolic Subgroups

The total phenolic content in samples was determined by the Folin–Ciocalteu method [24] and the results are expressed as mg of gallic acid equivalents per litre of extract (mg GAE/L).

Total flavonoids were determined using the colorimetric method reported by Yang et al. [25]. The results are expressed as mg of quercetin equivalents per litre of extract (mg QE/L).

Tannins were detected using vanillin-HCl according to the procedure described by Julkunen-Titto [26]. In this assay, catechin was used as the standard and the results are expressed as mg catechin equivalents per litre of extract (mg CE/L).

#### 2.3.2. HPLC Analysis of Individual Phenolics

For the separation, quantification and identification of individual phenolics, the HPLC method was used. The method was described in the study of Generalić Mekinić et al. [27]. The flow rate was 0.8 mL/min and the signal was monitored at 280 nm. The following solvents were used: solvent A was water/phosphoric acid (99.8:0.2, *v*/*v*) and solvent B was methanol/acetonitrile (50:50, *v*/*v*). The detected phenolic acids were identified by comparing their retention times and absorption spectra with those acquired for corresponding standards and by sample spiking. The compounds were quantified using external standard calibration curves. Phloroglucinol was detected using the same method but the signal was monitored at 267 nm, where the maximal absorption spectrum of this compound has been recorded. The results are expressed as mg of compound per litre of extract (mg/L).

### 2.4. Antioxidant Activity

#### 2.4.1. Ferric Reducing/Antioxidant Power (FRAP)

The reducing activity of the samples was measured as FRAP value according to the procedure reported by Benzie and Strain [28]. This method is used to measure the ability of samples to reduce ferric-tripyridyltriazine (Fe^3 +^-TPTZ) to a ferrous-tripyridyltriazine complex (Fe^2 +^-TPTZ) at a low pH value (3.6). Diluted extracts (10 µL) were added to a freshly prepared FRAP reagent (300 mM acetate buffer:TPTZ in 40 mM HCl:20 mM FeCl_3_ × 6H_2_O = 10:1:1) (300 µL) and the absorbances were measured at 593 nm. The results obtained by this method are expressed in millimoles of Trolox equivalents per litre (mM TE) [25].

#### 2.4.2. Radical 2,2-Diphenyl-1-picrylhydrazyl (DPPH)

The free radical scavenging activity against DPPH· was determined according to the procedure described by Katalinić et al. [29]. The decrease in absorbance of the initial DPPH solution (300 µL) after addition of the sample (10 µL) was monitored at 517 nm after 1 h. The results of the “quenching” reactions of antioxidants with DPPH are expressed in micromoles of Trolox equivalents per litre (µM TE). 

#### 2.4.3. Oxygen Radical Antioxidant Capacity (ORAC)

An ORAC assay was performed according to the procedure described by Generalić Mekinić et al. [27]. For the measurements, 25 µL of the diluted sample/phosphate buffer (blank)/standard (Trolox) was added to a well of a black 96-well plate. After that, 150 µL of the fluorescein was added, and after incubation for 30 min, 25 µL of the 2,2′-azobis(2-methylpropionamidine) dihydrochloride solution was added to initiate the reaction. The reaction was monitored every minute for 80 min, and the results are expressed as mM TE.

### 2.5. Statistical Analysis

For data analysis, STATISTICA (Data Analysis Software System, v. 13, StatSoft Inc., Tulsa, OK, USA) was used. Pearson’s correlation coefficient and principal component analysis (PCA) were used for determining the relations between the variables. All data are expressed as mean ± standard deviation (SD).

## 3. Results and Discussion

Phenolic compounds are plant secondary metabolites, and the isolation, characterisation and investigation of their biological activities have been the aim of much research. Generally, phenolics, but also the subgroup of phlorotannins, which are the dominant brown algal phenolics, are proven to have positive pharmacological and nutraceutical properties [2,12,14]. These health-promoting activities of phenolics are critical for neutralising the effects of oxidizing agents (such as free radicals) that are generated during metabolism due to exposure to extreme conditions (e.g., UV radiation, salinity, temperature, high oxygen concentrations, etc.). The concentration of phenolics in brown algae is influenced by different abiotic and biotic factors, while their analysis is influenced by their chemical nature, the applied extraction procedure, storage conditions, the presence of interfering substances, etc. [5,30]. In this study, different extraction modes were used to obtain the most effective algae extracts with the highest share of phenolics. Recent trends in algal research have also focused on the investigation and application of innovative technologies in improving the extraction efficiencies, and ultrasound-assisted extraction is a low-cost method that is often used [5,18,31,32]. 

As it is well known that the extraction rate may be improved by the modification of process variables, the ultrasound-assisted extractions of algal samples were performed at different temperatures (20, 40 and 60 °C) and using different solvents (water and ethanol). The compound chemical nature is a restricting factor in finding a suitable extraction solvent system [5]. Although researchers use various organic solvents in order to obtain extracts with a high share of phenolics [3,27,28,29,30,32], water and ethanol are preferred in the food, pharmaceutical and cosmetic industries due to economic, toxicological and environmental reasons [33]. Due to variations in the polarity of the phenolic compounds, it is expected that hydroalcoholic mixtures could be the most suitable solvents. This is confirmed by the results of this study, as a better extraction yield was obtained using alcoholic solvents in contrast to water solvents. Figure 1 shows the total contents of phenolics, flavonoids and tannins in extracts of *D. dichotoma* and *P. pavonica*.

The total phenolic content ranged from 127 mg GAE/L in the water extract of *D. dichotoma* prepared at 40 °C, to 423 mg GAE/L in the ethanolic extract of *P. pavonica*. Although the use of high temperatures usually leads to a kinetic improvement, it is often limited by the fact that most phenolics are not thermostable, so heat treatments could reduce the total extracted amount [34]. Garcia-Vaquero et al. [31] also reported that ultrasound-assisted extraction at 40 °C (for 30 min) resulted in phenolic extracts with the highest yield and maximal antioxidant capacity. As can be seen from the obtained results, the extracts prepared at RT in all cases, except for ethanolic extracts of *P. pavonica*, yielded the highest amount. The share of phenolics in the ethanolic extracts of *D. dichotoma* prepared at RT was 29% higher than that in the water extracts, and 56% and 46% higher in the ethanolic extracts prepared at 40 and 60 °C, respectively. Although there were no significant differences between the phenolic potential of the *D. dichotoma* and *P. pavonica* water extracts, the content of phenolics in the ethanolic extracts was significantly higher. Other authors also investigated phenolics from *D. dichotoma* [2,23,34,35,36] and *P. pavonica* [3,22,23,33,34], but comparison of the results is difficult due to the employment of different extraction protocols (solvents used, time of extraction, temperature, additional actions such as stirring, use of novel techniques, etc.) or due to results’ expression using different standard compounds. However, the results of our study on the influence of drying and extraction methods on *P. pavonica* phenolics showed better extraction yields in water extracts (more than twofold higher results were obtained) [23]. Figure 1b also shows the distribution of flavonoids among samples. It can be seen that the ethanolic extracts are richer in these valuable compounds, especially extracts of *D. dichotoma* (concentration range from 871 mg QE/L in extract prepared at 60 °C to 975 mg QE/L in extract prepared at RT). Kosanic et al. [22] also reported a higher content of flavonoids in extracts of *D. dichotoma* than in *P. pavonica,* while Čagalj et al. [23] obtained a higher content of flavonoids in ethanolic extracts. All other samples contained significantly lower amounts, from 23 to 160 mg QE/L. Similar results were obtained for the total tannins (Figure 1c), where, again, the highest amounts of these compounds were detected in the ethanolic extracts of *D. dichotoma* (from 0.34 to 0.39 mg CE/L).

The individual phenolic acids in the *D. dichotoma* and *P. pavonica* extracts detected by HPLC are presented in Table 1, while the phloroglucinol content is shown in Figure 1d. According to the presented results, the dominant phenolic acid in the extracts of *D. dichotoma* was *trans*-ferulic acid, with the highest concentrations in the ethanolic extract prepared at RT. In *P. pavonica*, the dominant phenolic was protocatechuic acid. Although *trans*-ferulic acid was also found in the *P. pavonica* extracts, its concentration was only significant in the water extract prepared at RT (1.22 mg/L). While the levels of *o*-coumaric acid in the *P. pavonica* extracts were low in all samples, the concentrations of its *para*- isomer were significantly higher in the ethanolic extracts than in the water extracts. Similar results were obtained in the *D. dichotoma* extract, where the highest concentration was detected in the ethanolic extract prepared at 20 °C (2.07 mg/L). Generally, the concentrations of hydroxycinnamic acid derivatives (*p*-coumaric, *o*-coumaric and *t*-ferulic acid) were higher in the *D. dichotoma* extracts, while the *P. pavonica* extracts were richer in hydroxybenzoic (protocatechuic and *p*-hydroxybenzoic) acids. According to the results for phloroglucinol content (Figure 1d), it is apparent that the *D. dichotoma* extracts were richer in this substance, with an almost threefold higher content in the water extracts than in the EtOH extracts. On the other hand, the *P. pavonica* ethanolic extracts contained significantly lower amounts of phloroglucinol, while its quantification in the water extracts was not possible (it was present, but at an amount below the quantification limit).

Due to the presence of phlorotannins, it has been reported that brown algae species possess higher antioxidant activity than green and red algae do. It is well known that the concentrations of these compounds vary according to numerous factors, such as species, season, age, geographical location and environmental conditions [7,9]. In this study, antioxidant activity was evaluated by means of a multiple-method approach, using three assays: Ferric Reducing/Antioxidant Power (FRAP), scavenging of the stabile 2,2-diphenyl-1-picrylhydrazyl (DPPH) radical and Oxygen Radical Antioxidant Capacity (ORAC). The obtained results are presented in Table 2. According to the presented results for the reducing activity of the samples, the highest activities were detected for the ethanolic extracts of *D. dichotoma* (from 690 to 792 mM TE), while all other samples had more than threefold lower activity. Although in most cases, the antioxidant activity of the samples shows a high correlation with phenolic content, this study only confirmed the significant impact of the flavonoid content on the reducing activity of the extracts (*r* = 0.9873, *p* < 0.0001).

Generally, very low DPPH activity of the samples was detected, with the ethanolic extracts of *P. pavonica* being the most active (from 501 to 645 µM TE). Similar results were obtained in the studies of Kosanić et al. [22] and Khaled et al. [35], where the free radical scavenging activities of all three investigated algae species (*D. dichotoma*, *P. pavonica* and *Sargassum vulgare*) were low (IC_50_ values significantly higher in comparison to ascorbic acid, BHA and α-tocopherol). 

Finally, for antioxidant activity measurements, an ORAC assay was also used. This method reflects classical radical chain breaking activity and measures inhibition of peroxyl-radical-induced oxidation [37]. Again, as can be seen in Table 2, ethanolic extracts were superior in comparison with the water extracts of both algae species. While the activity of the *P. pavonica* ethanolic extracts was about twofold higher than the activity of the water extracts, the ethanolic extracts of *D. dichotoma* showed excellent activity. In order to obtain results, these extracts were previously diluted to 1:400. 

In order to describe and identify the similarities and differences among the samples, a principal component analysis (PCA) was used. The correlation loadings of the first two principal components (PCs) shown in Figure 2 suggest high correlations of the studied parameters. The variables of flavonoids and tannins were strongly characterised by PC1 and showed a high mutual correlation (*r* = 0.9752, *p* < 0.001). Furthermore, those parameters that showed similar characteristics with the FRAP value and *p*-coumaric content in the extracts positively correlated (Figure 2a). The total phenolic content and the free radical scavenging assays (DPPH and ORAC) were characterised by PC2. The first two PCs described 86.81% of the initial data variability. The score plot (Figure 2b) shows the position of the measured parameters in the multivariate space of the first two PCs. The clear separation between samples indicates the differences between the solvent used and not the applied temperature during the extraction. The water extracts of both algae are grouped in the upper part of the plot, showing no significant difference between them, and are characterised by a low content of phenolics. Based on the contents of flavonoids and tannins, the ethanolic extracts were separated into two opposite parts and, due to the higher phenolics content, are positioned in lower part of the plot. 

The obtained results from the PCA are in accordance with the Pearson product moment correlations between variables that showed a high correlation between total phenolics and both FRAP (*r* = 0.8005, *p* = 0.0018) and ORAC values (*r* = 0.9483, *p* < 0.001), FRAP and phlorotannin content (*r* = 0.8951, *p* = 0.001), flavonoids and tannins (*r* = 0.9744, *p* < 0.001), flavonoids and reducing activity (*r* = 0.9739, *p* < 0.001) and DPPH and ORAC (*r* = 0.8033, *p* = 0.0017).

## 4. Conclusions

Brown algae are a valuable source of biologically active compounds. Research on the phenolic profile of the Adriatic algae *D. dichotoma* and *P. pavonica* is important for the interpretation of their biological potential and, therefore, for their potential use in the food, cosmetic and/or pharmaceutical industries. The results of this study also confirmed that the investigated algal species are a rich source of phenolics. The ethanolic extracts contained higher concentrations of phenolics, while the extraction temperature did not influence the extraction yield. *D. dichtoma* contained higher concentrations of hydroxycinnamic acid derivatives, while *P. pavonica* were richer in hydroxybenzoic acids. The tested extracts also showed good antioxidant potential using all three antioxidant assays, with flavonoids and tannins probably being responsible for this activity. This is still a relatively new scientific area, and further research should be directed toward the investigation of other extraction parameters or novel technologies focused on the yield of extracted bioactive compounds, especially phlorotannins. Furthermore, it would be interesting to investigate the other biological properties of brown algae extracts.

## Figures and Tables

**Figure 1 foods-10-01187-f001:**
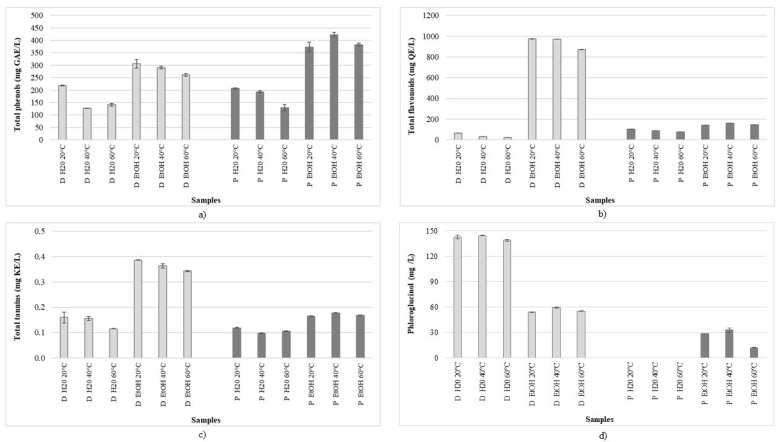
Total contents of (**a**) phenolics, (**b**) flavonoids, (**c**) tannins and (**d**) phloroglucinol in *Dictyota dichotoma* (D) and *Padina pavonica* (P) extracts.

**Figure 2 foods-10-01187-f002:**
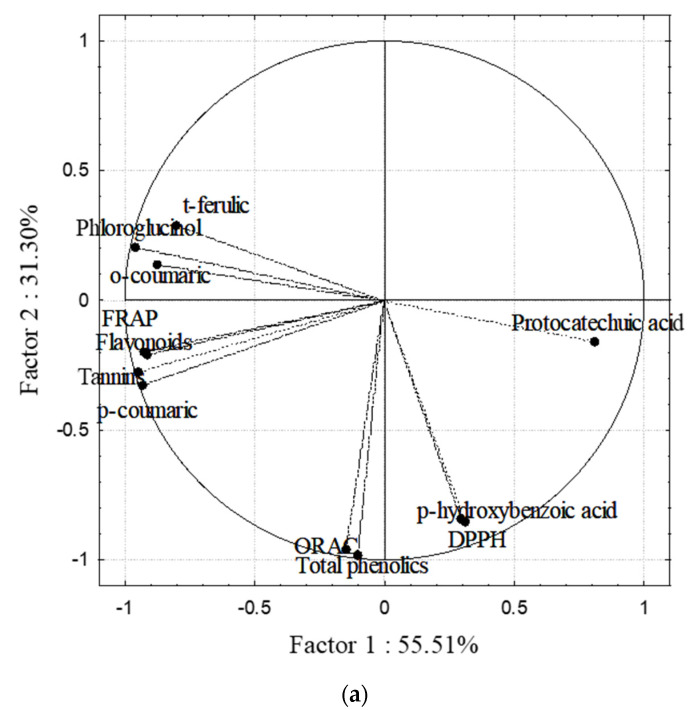
Correlation of the (**a**) loading plot and (**b**) score plot of the PCA.

**Table 1 foods-10-01187-t001:** Phenolic acids in *Dictyota dichotoma* (D) and *Padina pavonica* (P) extracts detected by HPLC.

Sample	Compound Concentration (mg/L of Algal Extract)					
	Protocatechuic Acid	*p*-hydroxybenzoic Acid	*p*-coumaric Acid	*t*-ferulic Acid	*o*-coumaric Acid	*∑*
D H_2_O 20 °C	0.62 ± 0.05	0.22 ± 0.04	1.00 ± 0.02	1.41 ± 0.15	0.16 ± 0.01	3.41
D H_2_O 40 °C	0.05 ± 0.01	0.19 ± 0.02	0.72 ± 0.01	1.28 ± 0.10	0.10 ± 0.01	2.34
D H_2_O 60 °C	0.30 ± 0.00	0.13 ± 0.01	0.44 ± 0.02	0.68 ± 0.02	0.02 ± 0.00	1.57
D EtOH 20 °C	0.05 ± 0.00	0.42 ± 0.05	2.07 ± 0.20	1.57 ± 0.07	0.17 ± 0.01	4.29
D EtOH 40 °C	0.23 ± 0.04	0.30 ± 0.01	1.75 ± 0.15	1.37 ± 0.01	0.14 ± 0.00	3.78
D EtOH 60 °C	0.14 ± 0.00	0.32 ± 0.06	1.77 ± 0.01	1.33 ± 0.03	0.12 ± 0.03	3.66
P H_2_O 20 °C	1.68 ± 0.03	0.51 ± 0.56	0.03 ± 0.00	1.22 ± 0.10	0.02 ± 0.00	3.46
P H_2_O 40 °C	1.70 ± 0.02	0.60 ± 0.02	0.02 ± 0.00	0.07 ± 0.00	0.03 ± 0.00	2.42
P H_2_O 60 °C	1.42 ± 0.10	0.60 ± 0.01	0.02 ± 0.00	0.07 ± 0.01	0.03 ± 0.00	1.55
P EtOH 20 °C	1.34 ± 0.03	0.75 ± 0.02	0.69 ± 0.07	0.21 ± 0.01	0.01 ± 0.01	3.00
P EtOH 40 °C	1.05 ± 0.09	0.76 ± 0.03	0.79 ± 0.04	0.24 ± 0.05	0.02 ± 0.00	2.86
P EtOH 60 °C	1.15 ± 0.08	0.66 ± 0.07	0.88 ± 0.16	0.33 ± 0.05	0.03 ± 0.00	3.06

**Table 2 foods-10-01187-t002:** Antioxidant activity of *Dictyota dichotoma* (D) and *Padina pavonica* (P) extracts detected by the FRAP, DPPH and ORAC methods.

Sample	FRAP (mM TE)	DPPH (µM TE)	ORAC (mM TE)
D H_2_O 20 °C	139 ± 16	196 ± 47	26.7 ± 2.4
D H_2_O 40 °C	147 ± 3	144 ± 27	26.6 ± 1.5
D H_2_O 60 °C	232 ± 7	125 ± 36	20.6 ± 4.5
D EtOH 20 °C	792 ± 9	113 ± 31	39.1 ± 0.8 *
D EtOH 40 °C	717 ± 17	84 ± 6	40.8 ± 2.4 *
D EtOH 60 °C	690 ± 21	109 ± 9	38.7 ± 2.3 *
P H_2_O 20 °C	187 ± 7	30 ± 10	25.7 ± 1.2
P H_2_O 40 °C	147 ± 8	84 ± 6	22.5 ± 1.9
P H_2_O 60 °C	106 ± 9	56 ± 11	26.2 ± 0.8
P EtOH 20 °C	202 ± 8	501 ± 66	46.2 ± 1.4
P EtOH 40 °C	231 ± 12	595 ± 13	55.8 ± 3.0
P EtOH 60 °C	214 ± 24	645 ± 25	51.2 ± 0.7

* The result obtained for samples deleted 400-times.

## Data Availability

Data are available on request.

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
