# Peer review of "Bioactive Phenolic Metabolites from Adriatic Brown Algae Dictyota dichotoma and Padina pavonica (Dictyotaceae)"

_foods, 2021, doi:10.3390/foods10061187_

Round 1
Reviewer 1 Report
Dear Authors
The present manuscript is well organised and presented, although there are some opportunities for further improvement. Please find here my suggestions.
1- Ferric Reducing/Antioxidant Power (FRAP) , Radical 2,2-diphenyl-2-picrylhydrazyl (DPPH) and Oxygen Radical Antioxidant Capacity (ORAC) can be explained in more details in materials method section. It may be easier for readers to reproduce the methodology.
2- There is lack of recent references in discussion, only one reference from 2020-2021 has been considered in the paper.
Thank you
Regards
Reviewer 2 Report
Introduction, first paragraph, sentence is too long, break it two.
Lines 66-67: “Beside brown algae make contribute in food production as valuable ingredient in formulation of functional food.” English not good, rewrite, do not start with “Beside”. Cite appropriate reference.
Lines 68-69: “In the past years the interest in seaweeds has markedly increased and number of research studies on their metabolites with biological activities have been conducted so” repetition, remove it.
As it has been stated by the authors brown algae has been studied extensively, then what is new and important about Dictyota dichotoma and Padina pavonica available at Adriatic Sea? What is the pressing reason? Was there any dearth of information about these two species?
Lines 73-75:” Furthermore, the aim of this study was to found the connection between algae phenolic profile and related biological activity in order to find the most suitable extraction parameters for the production of bioactive extracts with potential use in food industry.” This aim is convincing. However, potential role of phenolic compounds toward their biological activities has long been established, furthermore there are several reports on the extraction methods as well. In that regard, what would be novelty this manuscript could offer to the readers?
Overall, introduction needs to be revised with emphasis on highlighting the novel and pressing scientific reasons.
Line 175: it should read as “obtain”
Line 177: replace “to be used for preparations of extracts that are planned to be used in” with “for”
Line 178, replace “or” with “and”
Line 192: replace “phenolic content in extracts” with “total extracted amount”
Line 194: replace “contained” with “yielded”, and remove “of phenolics”
Line 196: remove “prepared by the same protocol”
Lines 197-198: remove “in comparison to the water extracts pre- 197 pared at same temperatures.”
Line 274: “positively correlated” might be better.
Line 291: “Brown algae are potentially new and valuable source of biologically active compounds”, not really so, Brown algae is being researched for a long time, authors started in the introduction too.
Overall, sentence construction needs improvement, at many places repetition of concepts is noticed in the same sentence with long long sentences.
Lines 294-298: “This is still a relatively new scientific area but further researches should be directed to the investigation of some other extraction parameters or novel technologies on yield of extracted bioactive compounds, especially phlorotannins, as well as on investigation of other biological properties of brown algae extracts.” Too long to comprehend, divide into 2-3 sentences.
Instead, “Conclusions” could be replaced with “Summary” and include the research carried out in this submission with some results.
Reviewer 3 Report
Extraction of phenols with ethanol and water at different temperatures for two Adriatic algae.
The topic is now very popular. It is interesting because algae could become very important for food/nutraceuticals and pharmacological interest.
The algae uses are original and much studied in recent years, on the other hand the methods used for the investigations are already well known and used.
It adds some indication of general composition of the two algae, which could be useful for other researchers researchers as a starting point to then use innovative extractive methods and also more sensitive instruments such as LC/MS-MS.
The conclusions are somewhat generic. In the conclusions I would summarize in two lines the advantages of the two types of extract to give emphasis to the work, which certainly is basic for other researchers to broaden the subject matter.
Precisely I would express the DPPH results as in micromoles of Trolox equivalents per litre (μmol TE/L) as the other antioxidant tests utilized (ORAC and FRAP), or alternatively as IC50 value of DPPH assay.
In the conclusions highlight the differences between extraction solvents and temperature
Round 2
Reviewer 2 Report
Revised draft is reading well.